# Jasmonic Acid Signaling Pathway in Response to Abiotic Stresses in Plants

**DOI:** 10.3390/ijms21020621

**Published:** 2020-01-17

**Authors:** Md. Sarafat Ali, Kwang-Hyun Baek

**Affiliations:** Department of Biotechnology, Yeungnam University, Gyeongsan, Gyeongbuk 38541, Korea; sarafatbiotech@ynu.ac.kr

**Keywords:** abiotic stresses, jasmonates, JA-Ile, JAZ repressors, transcription factor, signaling

## Abstract

Plants as immovable organisms sense the stressors in their environment and respond to them by means of dedicated stress response pathways. In response to stress, jasmonates (jasmonic acid, its precursors and derivatives), a class of polyunsaturated fatty acid-derived phytohormones, play crucial roles in several biotic and abiotic stresses. As the major immunity hormone, jasmonates participate in numerous signal transduction pathways, including those of gene networks, regulatory proteins, signaling intermediates, and proteins, enzymes, and molecules that act to protect cells from the toxic effects of abiotic stresses. As cellular hubs for integrating informational cues from the environment, jasmonates play significant roles in alleviating salt stress, drought stress, heavy metal toxicity, micronutrient toxicity, freezing stress, ozone stress, CO_2_ stress, and light stress. Besides these, jasmonates are involved in several developmental and physiological processes throughout the plant life. In this review, we discuss the biosynthesis and signal transduction pathways of the JAs and the roles of these molecules in the plant responses to abiotic stresses.

## 1. Introduction

Plants grow in environments that impose a variety of biotic and abiotic stresses. The primary abiotic stresses that influence plant growth include light, temperature, salt, carbon dioxide, water, ozone, and soil nutrient content and availability [1], where the fluctuation of any of these can hamper the normal physiological processes. Being static organisms, plants are unable to avoid abiotic stresses simply by moving to a suitable environment. Consequently, they have evolved mechanisms to compensate for the unwanted stressful conditions by altering their own developmental and physiological processes.

The growth, development, and survival of plants depend on complex biological networks coupled with anabolic and catabolic pathways [2]. Abiotic stresses can disrupt these network pathways, resulting in their uncoupling. For example, extremely high or low temperatures might inhibit a subset of enzymes in the same or connected pathways [3], and hence various intermediate compounds might accumulate as a result of this functional uncoupling of metabolic pathways [4]. These intermediate compounds could be converted to toxic by-products that might affect the cell’s survival or longevity [5]. Reactive oxygen species (ROS) are one of the most common groups of toxic intermediates produced by abiotic stresses.

Phytohormones, the regulators of plant development, are central players in sensing and signaling diverse environmental conditions, such as drought, osmotic stress, chilling injury, heavy metal toxicity, etc. [6]. There are currently nine known major classes of naturally occurring phytohormones (viz., auxins, gibberellins, cytokinins, abscisic acid (ABA), ethylene (ET), brassinosteroids, jasmonic acid (JA), salicylic acid (SA), and strigolactones), all of which evoke many different responses.

Specifically, JA and its derivatives (e.g., jasmonyl isoleucine (JA-Ile), *cis*-jasmone, JA-glucosyl ester, methyl jasmonate (MeJA), jasmonoyl-amino acid, 12-hydroxyjasmonic acid sulfate (12-HSO_4_-JA), 12-*O*-glucosyl-JA, JA-Ile methyl ester, JA-Ile glucosyl ester, 12-carboxy-JA-IIe, 12-*O*-glucosyl-JA-IIe, and lactones of 12-hydroxy-JA-IIe), which are collectively known as jasmonates (JAs), are fatty acids derived from cyclopentanones and belong to the family of oxidized lipids that are collectively known as oxylipins [7]. These oxylipins are biologically active signaling molecules that are produced either enzymatically by lipoxygenases or alpha-dioxygenases, or nonenzymatically through the autoxidation of polyunsaturated fatty acids [8].

The JAs are ubiquitous in higher plant species, where their levels are high in the reproductive tissues and flowers, but very low in the mature leaves and roots [9,10]. JAs modulate many crucial processes in plant growth and development, such as vegetative growth, cell cycle regulation, anthocyanin biosynthesis, stamen and trichome development, fruit ripening, senescence, rubisco biosynthesis inhibition, stomatal opening, nitrogen and phosphorus uptake, and glucose transport [10,11,12,13,14,15,16,17,18,19,20,21,22,23,24,25]. As signaling molecules, JAs regulate the expression of numerous genes in response to abiotic stresses (e.g., salt, drought, heavy metals, micronutrient toxicity, low temperature, etc.) and promote specific protective mechanisms (Figure 1) [26]. In this review, we focus on the biosynthesis and signaling of JA, *cis*-jasmone, MeJA, and JA-Ile in response to abiotic stresses because of the high bioactivity of these compounds.

## 2. Abiotic Stress-Sensing Mechanisms in Plants

Abiotic stresses alter the physiological processes in plants by affecting gene expression, RNA or protein stability, the coupling of reactions, ion transport, or other cellular functions [27]. Any of these alterations could be a signal to the plant that a change in environmental conditions has occurred and that it is the optimum time to respond by either activating the stress-response pathways or altering existing ones. Some of the mechanisms used by plants to sense the abiotic stresses are as follows [28]: (i) Physical sensing, involving mechanical effects of the stress on the plant or cell structure, such as contraction of the plasma membrane from the cell wall during drought stress; (ii) biophysical sensing, involving changes of the protein structure or enzymatic activity, such as the inhibition of different enzymes during heat stress; (iii) metabolic sensing, involving the detection of by-product accumulation due to the uncoupling of electron transfer or enzymatic reactions, such ROS accumulation due to high light intensity; (iv) biochemical sensing, involving the presence of specialized proteins to sense a particular stress, such as calcium channels that can alter the Ca^2+^ homeostasis and sense changes in the temperature; and (v) epigenetic sensing, involving modifications of the DNA or RNA structure without altering the genetic sequences, such as the changes in chromatin that occur during temperature stress [28,29,30]. These stress-sensing mechanisms can activate downstream signal transduction pathways individually or in combination. Consequently, plants activate various anti-stress mechanisms to acclimate or adapt to the various stresses.

## 3. Biosynthesis and Metabolism of Jasmonic Acid during Abiotic Stress

During the last decades, the biosynthesis of JA has been well characterized in a variety of monocotyledonous and dicotyledonous plants [10,31,32]. To summarize, JA is biosynthesized through the consecutive action of enzymes present in the plastid, peroxisome, and cytoplasm (Figure 2) [33]. Abiotic (and biotic) stimuli activate phospholipases in the plastid membrane, promoting the synthesis of linolenic acid (18:3) in the plant [10,34]. Linolenic acid, a precursor in the JA biosynthesis process, is converted to 12-oxo-phytodienoic acid (12-oxo-PDA) through oxygenation with lipoxygenase (LOX), allene oxide synthase (AOS), and allene oxide cyclase (AOC). JA is then synthesized from 12-oxo-PDA by the activity of 12-oxo-phytodienoic acid reductase (OPR) and 3 cycles of beta-oxidation. Therefore, the JA biosynthetic pathway is known as the octadecanoid pathway [32,34,35].

In the cytosol, JA metabolic pathways convert the phytohormone into more than 30 distinct active and inactive derivatives, depending on the chemical modification of the carboxylic acid group, the pentenyl side chain, or the pentanone ring (Figure 2) [36,37,38,39,40]. Among the series of metabolites, free JA, *cis*-jasmone, MeJA, and JA-Ile are considered to be the major forms of bioactive JA in plants [10,41]. *cis*-jasmone is produced through the decar-boxylation of JA (Figure 3) [42]. The volatile MeJA is produced from JA through the activity of JA carboxyl methyltransferase (Figure 3) [26]. Jasmonate amino acid synthetase 1 (JAR1) catalyzes the reversible conversion between JA and JA-Ile (Figure 3) [41]. Evidence suggests that JA-Ile is an important compound in the JA signal transduction pathway [43].

## 4. Jasmonic Acid Signaling during Abiotic Stress

In the plant cell cytoplasm, the most bioactive JA is JA-Ile, the level of which is very low under normal conditions [41]. Upon stress stimulation, JA undergoes epimerization to form JA-Ile, which accumulates in the cytoplasm of the stressed leaves. JA-Ile is transported to the nucleus and adjacent sites of the leaves for defensive responses [44,45]. In *Arabidopsis thaliana* (At), the subcellular localization of JAs are regulated by a high-affinity transporter, jasmonic acid transfer protein 1 (AtJAT1, also known as AtABCG16) [46]. Both the plasma membrane and nuclear membrane of plant cells contain JAT1, through which JA or JA-Ile is exported from the cytoplasm to the nucleus and apoplast [46]. Therefore, the dynamics of JA or JA-Ile in the cytoplasm, nucleus, and apoplast is regulated by JAT1 during abiotic stress.

JA or JA-Ile in the apoplast activates the JA signaling pathways in other cells. JA signals can transmit long distances via vascular bundles and/or air transmission. After their synthesis, JA and MeJA are transmitted in plants systemically [47]; that is, they can transfer to different parts of the plant via the vascular bundles [48]. During such transportation, JAs are not only transported but are also resynthesized [47], a fact that has been proven by the localization of various JA synthetases in the companion cell–sieve element complex of the vascular bundles in the tomato plant [49]. The JA precursor 12-oxo-PDA is formed in the sieve elements of the phloem, which is another indication of the resynthesis of JAs transported through the vascular bundles [50]. Compared with JA, MeJA can diffuse easily to distant leaves and adjacent plants owing its strong volatility and high capability of penetrating the cell membrane [40].

Under normal conditions, the promoters of jasmonate-responsive genes are not activated by the different types of transcription factors (TFs) due to the low level of JA-Ile (Figure 4). The various TFs [51] are repressed by a series of jasmonate-zinc finger inflorescence meristem (ZIM) domain (JAZ) proteins that act as transcriptional repressors (Table 1). The JAZ repressors recruit the protein topless (TPL) and the interactor/adaptor protein novel interactor of JAZ (NINJA); together, they form an effective transcriptional repression complex that acts to inhibit the expression of jasmonate-responsive genes by changing the open complex to a closed one through the further recruitment of histone deacetylase 6 (HDA6) and HDA19 [43,52,53,54,55].

To date, 13 JAZ proteins have been identified in *Arabidopsis*, most of which have two conserved domains: the central domain known as the ZIM domain [56,57,58,59], and the C-terminal JA-associated (Jas) domain [56]. The various domains present in the JAZ proteins facilitate their protein-protein interactions [60]. The JAZ proteins interact with the TFs via the ZIM domain, interacting with NINJA (which contains an ethylene-responsive element binding factor-associated amphiphilic repression (EAR) motif) and recruiting TPL to form the JAZ–NINJA–TPL repressor complex [54,55]. Among the 13 JAZ proteins of *Arabidopsis*, JAZ5, JAZ6, JAZ7, JAZ8, and JAZ13 contain an additional EAR motif that can interact directly with TPL in the absence of NINJA [57,59]. Within the Jas domain, the minimal amino acid sequence that can bind the coronatine or JA-Ile is termed the JAZ degron, the bipartite structure of which contains a loop and an amphipathic alpha-helix that binds to coronatine or JA-Ile and coronatine insensitive 1 (COI1), respectively [61].

Abiotic stresses elevate the processes that lead to JA-Ile formation in the cytosol and its transportation to the nucleus. JA-Ile is the natural bioactive ligand of *A. thaliana*, as affirmed by gas chromatography-mass spectrometry and high-performance liquid chromatography analyses [41]. Among JA, JA-Ile, MeJA, and 12-oxo-PDA, only JA-Ile can promote *COI1*-JAZ binding [58].

The ubiquitin–proteasome complex comprises suppressor of kinetochore protein 1 (SKP1)–cullin–F-box (SCF). The *Arabidopsis COI1* mutant lacks all responses to JA [62]. The *COI1* gene encodes an F-box protein, which associates with SKP1 and cullin to form SCF-type E3 ubiquitin ligase [63]. During abiotic stress, the JA-Ile that is formed and transported to the nucleus is recognized by the F-box protein COI1. JA-Ile facilitates the interaction of JAZ with COI1 within the SCF complex [63,64], with inositol pentakisphosphate serving as a cofactor in the formation of the COI1–JAZ co-receptor complex [61,65]. Ubiquitination of the JAZ protein leads to its proteasomal degradation and the release of the TFs to modulate the expression of jasmonate-responsive genes, thereby regulating the jasmonate-regulated defenses and growth. Mediator 25 (MED25), a subunit of the *Arabidopsis* mediator complex [66], bridges the communication between the gene-specific TF, RNA polymerase II, and the general transcription machinery [67]. Several lines of evidence have indicated that every aspect of JA function is due to the matching pairs of TFs with a subset of JAZ repressors to orchestrate the expression of jasmonate-responsive genes [64,68,69,70,71].

## 5. Regulation of Diverse Jasmonic Acid Responses by Transcription Factors during Abiotic Stress

Abiotic stresses induce JA signaling through the derepression of TFs. JAZ proteins interact with the MYC and MYB TFs and suppress the expression of jasmonate-responsive genes [56]. JAZ proteins are stimulated for proteosomal degradation in the presence of the bioactive ligand JA-Ile [56]. Studies have revealed that several other TFs (e.g., NAC, ERF, and WRKY) are also involved in JA signaling [87,88,89]. In addition to the TFs, JA signaling also activates the calcium channel [90], mitogen-activated protein kinase cascade [45], and various other processes that interact with SA, ABA, and ET to govern plant growth and development in response to abiotic stresses [91].

MYC2, encoded by the *JIN1* gene, is a basic helix-loop-helix (bHLH) TF and a key regulator of JA signaling. MYC2 binds to the G-box (CACGTG) and G-box-related hexamers [76,92,93,94,95], and can interact with most members of the JAZ repressors [76]. However, it is the only MYC subtype that is not the target of JAZ repressors. A number of other TFs can interact with JAZ repressors and remodel the JA signals into specific context-dependent responses (Table 1). MYC3 and MYC4 have similar DNA-binding specificity as MYC2 and can interact with JAZ proteins [76]. MYC5 (bHLH28), which is closely related to MYC2, is activated by the JAs and is required for stamen development and seed production [96,97]. Besides the MYC TFs, the JA-associated MYC2-like (JAM) proteins bHLH3/JAM3, bHLH13/JAM2, bHLH14, and bHLH17/JAM1 regulate JA-mediated anthocyanin accumu-lation, chlorophyll loss, root growth, resistance to bacterial pathogens, and leaf senescence [83,84,85,98]. Inducer of CBF expression 1 (ICE1) and ICE2, which are bHLH-type TFs, interact with JAZ4 and JAZ9 for the regulation of JA-dependent freezing tolerance [68]. Rice salt sensitive 3 (RSS3) interacts with JAZ9 and JAZ11 and non-R/B-like bHLH TFs, forming the RSS3–JAZ–bHLH complex that regulates the JA-mediated salt stress response [79].

The MYB TFs, which belong to the R2R3-MYB family, show considerable response to JA signaling. They control many processes in plants; for example, the synthesis of tryptophan and glucosinolates is regulated by MYB51 and MYB34, which also play an important role downstream of MYC2 [76]. A subset of JAZ proteins repress the transcriptional activities of MYB21 and MYB24 through their N-terminal R2R3 domain [71]. Evidence suggests that MYB21 and MYB24 are crucial factors for regulating stamen development and pollen maturation in *Arabidopsis* [71]. Anthocyanin accumulation and trichome initiation are positively regulated by MYB75 [70]. MYB21 and MYB24 also interact with MYC2, MYC3, MYC4, and MYC5 to form an MYC–MYB transcription complex that regulates stamen development [97].

The NAC family of TFs is also activated by JA signaling. For example, the JA signal-activated proteins ATAF1 and ATAF2 are involved in the development of plant resistance to salt stress, drought, and plant pathogens like *Botrytis cinerea* [99]. ATAF1 and ATAF2 also play crucial regulatory roles in the oxidative stress caused by abiotic stresses. The NAC TF ANAC019 and ANAC055 work downstream of MYC2 to regulate cell division, secondary cell wall synthesis, and seed germination [100].

The TFs ORCA2 and ORCA3 belong to the AP2/ERF-domain family activated by JA signaling and regulate the expression of genes related to monoterpenoid indole alkaloid biosynthesis [101]. ORA59 regulates the biosynthesis of hydroxycinnamic acid amides and acts as the integrator of JA and ET signals [26,102]. ORA47 is a crucial regulator in the positive jasmonate-responsive feedback loop owing to the activation of the JA biosynthesis gene *AOC2* [103]. Jasmonate-responsive AtERF3 and AtERF4 act as repressors to downregulate the expression of their respective target genes and interfere with the activity of other activators [104]. JAZ repressors cannot repress the activity of the TFs directly, indicating the existence of adaptors or co-repressors in the JA signaling pathway.

WRKY TFs play a critical regulatory role in confronting environmental stresses, as well as in plant development and senescence. In *Arabidopsis*, WRKY70 [105], WRKY22 [106], WRKY50 [107], WRKY57 [69], and WRKY89 [108], which are regulated by the JA signaling pathway, are particularly associated with plant defense functions. In the *Nicotiana attenuata*, WRKY3 and WRKY6 increase the levels of JA and JA-Ile by regulating the expression of jasmonate biosynthesis-related genes (*LOX*, *AOS*, *AOC,* and *OPR*) [109]. In the *Arabidopsis* plant, WRKY57 combines with JAZ4 and JAZ8 to regulate JA-induced leaf senescence [69].

Filamentous flower (FIL), a YABBY family TF, interacts with JAZ3 to regulate JA-mediated responses, such as chlorophyll loss and anthocyanin accumulation [78]. Trichome initiation and anthocyanin accumulation in plants are regulated by the WD-repeat–bHLH–MYB protein complexes. JAZ1, JAZ8, and JAZ11 interact with these complexes and repress their transcriptional activity, leading to the inhibition of anthocyanin accumulation and trichome initiation [70]. Plants biosynthesize JA-Ile in response to environmental cues and induce the degradation of the JAZ proteins, thereby freeing the WD-repeat–bHLH–MYB complexes and allowing them to regulate the expression of genes essential for anthocyanin accumulation and trichome initiation [70,78].

## 6. Roles of Jasmonic Acid in Alleviating Abiotic Stresses in Plants

### 6.1. Jasmonic Acid Signaling under Salt Stress

Salinity stress has both osmotic and cytotoxic effects on plant growth and development. The endogenous JA content was increased in *A. thaliana* [110], tomato (*Lycopersicon esculentum*) [111], and potato (*Solanum tuberosum*) [112] after salt treatment. Transcript profile analysis of stressed sweet potato revealed that during salt stress JA level was significantly increased to cope with the effect of salt stress [113]. The JA content increased immediately and persistently in the salt-sensitive plants, whereas the changes were not significant in the salt-tolerant ones [112]. Exogenous MeJA increased the tolerance of the black locust tree (*Robinia pseudoacacia*) to salt stress by increasing the activities of superoxide dismutase (SOD) and ascorbate peroxidase (APX) [108]. These finding were similar to those of Faghih et al. [114], who found that MeJA enhanced the activities of the APX, peroxidase (POD), and SOD enzymes. These lines of evidence suggest that JAs can alleviate salt stress by increasing the endogenous hormones and the antioxidative system.

### 6.2. Jasmonic Acid Signaling under Drought Stress

Drought stress or water deficit decreases turgor pressure, increases ion toxicity, and inhibits photosynthesis. It has been reported in several studies that JA signaling pathways are associated with the alleviation of drought stress. The increase in the endogenous JA content was rapid and transient in *A. thaliana* [21] and citrus (*Citrus paradisi* × *Poncirus trifoliata*) [115] immediately after drought stress, but the content decreased to the basal level with prolongation of the stress. MeJA treatment could improve the drought resistance in peanut (*Arachis hypogaea*) [116], rice (*Oryza sativa*) [117], soybean (*Glycine max*) [118], and broccoli (*Brassica oleracea*) plants [119]. The application of exogenous MeJA not only increased the total carbohydrate, polysaccharide, total soluble sugar, free amino acid, total proline, and protein contents, but also the activities of catalase (CAT), POD, and SOD in maize plants (*Zea mays*) [120]. In the broad bean (*Vicia faba*) and barley (*Hordeum vulgare*) plants, MeJA increased their abilities to resist drought by regulating stomatal closure [121,122]. MeJA also increased the drought resistance of cauliflower (*B. oleracea*) by activating the enzymatic (SOD, POD, CAT, APX, and glutathione reductase) and nonenzymatic (proline and soluble sugar) antioxidative systems [119]. Therefore, MeJA effectively improves the drought tolerance of plants by increasing the organic osmoprotectants and antioxidative enzyme activity [123].

### 6.3. Jasmonic Acid Signaling under Heavy Metals Toxicity

Heavy metals can mimic the essential mineral nutrients and generate ROS. Several studies have revealed that JA signaling pathways are associated with heavy metal toxicity. Exogenous MeJA could alleviate the cadmium-induced damage in soybean (*G. max*) [124], *A. thaliana* [125], European black nightshade (*Solanum nigrum*) [126], chili pepper (*Capsicum frutescens*) [127], and mangrove (*Kandelia obovata*) plants by increasing the activities of SOD, APX, and CAT. MeJA mitigated the toxicity of boron in the sweet wormwood (*Artemisia annua*) by reducing the amount of lipid peroxidation and stimulating the synthesis of antioxidative enzymes [128]. In *B. napus,* oxidative stress was minimized by MeJA through the induction of the expression of genes encoding antioxidants and secondary metabolites [129]. Therefore, the exogenous application of MeJA effectively alleviates heavy metal damage by increasing the levels of antioxidative enzyme activity and secondary metabolites.

### 6.4. Jasmonic Acid Signaling under Micronutrient Toxicity 

Several reports have suggested that JAs can protect plants from the effects of micronutrient toxicity. A high boron concentration is detrimental to plant growth and development [130,131] as reported in the apple (*Malus domestica*) root stock [132], wheat (*Triticum aestivum*) [133], barley (*H. vulgare*) [134], and tomato plants [135]. Treatment with exogenous MeJA could counter the boron toxicity in plants by activating the antioxidative defense enzymes (CAT, POD, and SOD) and inhibiting lipid peroxidation [9,128]. JAs also play a crucial role in plant defense responses against lead (Pb) stress. JA showed a reduction in Pb uptake and increased the growth of tomato plants when seeds were primed with JA [136].

### 6.5. Jasmonic Acid Signaling under Freezing Stress

Low temperature or cold stress causes extracellular ice crystal formation and cell dehydration. JA signaling plays a prominent role in the adaptation of plants to cold stress. The expression of the MYC TFs and several cold-responsive genes (*MaCBF1, MaCBF2, MaKIN2, MaCOR1, MaRD2, MaRD5,* etc.) was induced after the cold storage of bananas (*Musa acuminata*) [137]. MeJA could alleviate the cold stress in the tomato [138], loquat (*Eriobotrya japonica*) [139], pomegranate (*Punica granatum*) [140], mango (*Mangifera indica*) [141], guava (*Psidium guajava*) [142], cowpea (*Vigna sinensis*) plant [143], and peach (*Prunus persica*) [144] by increasing the synthesis of antioxidants and the activation of some defense compounds (e.g., phenolic compounds and heat shock proteins). These results suggest that JAs can mitigate cold injury through their promotion of the active defense compounds and the antioxidative system.

### 6.6. Jasmonic Acid Signaling under Ozone Stress

Ozone generates ROS that cause lesions and induce programmed cell death in plants. In wild-type *Arabidopsis*, the JA content was found to be significantly increased after ozone treatment [145]. The spread of programmed cell death caused by ozone could be inhibited by exogenous treatment with MeJA [145,146,147,148]. Moreover, the hybrid poplar (*Populus maximowizii* × *P. trichocarpa*) and tomato (*L. esculentum*) showed reduced sensitivity to ozone after exogenous MeJA treatment [145,149]. Elevated ozone activated the JA pathway in tomato plants which significantly up-regulated the emission rates of volatile compounds for the protection of plants from natural enemies [150].

### 6.7. Jasmonic Acid Signaling under Light Stress

Fewer reports are available about the effects of light and the JA signal on plant growth and development. In several studies, the JA signaling pathways in *Nicotiana* and *Brassica* species were initiated by the JA biosynthesis induced by UVB treatment, which increased the defensive mechanisms of the plants [151,152]. JA signaling had an effect on blue light-mediated light morphogenesis in *A. thaliana* and tomato (*L. esculentum*) [153,154] and on red light/far-red light-mediated photomorphogenesis in *A. thaliana* and rice (*O. sativa*) [152].

### 6.8. Jasmonic Acid Signaling under CO_2_ Stress

There are few reports about the JA signal transduction pathway in plants under CO_2_ stress, however, these reports have varied for various plant and insect species [155,156,157]. Ballhorn et al. reported that in lima bean (*Phaseolus lunatus*), the concentration of MeJA and cis-JA was increased at a high concentration of CO_2_ (500, 700, and 1000 ppm) [158]. An elevated level of CO_2_ (750 ppm) increased the defense mechanism of tomato plants against nematode by activating the JA- and SA-signaling pathway [159]. The elevated level of CO_2_ also increased the JA and main defense-related metabolites in tobacco but decreased in rice [157].

## 7. Roles of Jasmonic Acid in Plant Species other than Angiosperms

The information herein regarding the biosynthesis and activities of JA and its derivatives is related to angiosperms. Aside from the angiosperms, the bryophytes, lycophytes, fern (lycophytes and ferns/horsetails, together known as pteridophytes), and gymnosperms have all been shown to contain JA compounds, including the precursor 12-oxo-PDA. Among the multicellular sporophytes (consisting of bryophytes and vascular plants), bryophytes such as the moss (*Physcomitrella patens*) and the liverwort (*Marchantia polymorpha*) produce 12-oxo-PDA but not JA [160,161], suggesting that only the first half of the octadecanoid pathway in chloroplasts remains in the bryophytes. 

Among the vascular plants, lycophytes (seedless vascular plants) such as the spikemoss (*Selaginella moellendorffii*) have been shown to possess 12-oxo-PDA, JA, and JA-Ile, and the endogenous concentrations of 12-oxo-PDA and JA were also transiently increased within 10 min after wounding [162]. Therefore, the evolution of the JA biosynthetic pathway after that of 12-oxo-PDA is related to the plant acquisition of a vascular system. JA biosynthesis and its signal transduction pathway were also observed in the fern (*Pteridium aquilinum*), where wounding stimulated 12-oxo-PDA and JA in the plant [163], suggesting that JA and JA-Ile biosynthesis first emerged after the emergence of the bryophytes in plant evolution.

Jasmonates also act as cellular signaling compounds in gymnosperms [164,165]. As shown in several studies, the application of MeJA increased the resistance of the Norway spruce (*Picea abies*) to the root pathogen *Pythium ultimum* Trow [166], induced the expression of the 14-3-3 gene in the spruce plant [*Picea glauca* (Muench) Voss] [167], and accumulated a high amount of paclitaxel in several *Taxus* species [168]. The accumulation of JA in response to wounding is a common physiological feedback among all vascular plant species [1]. Therefore, JA has evolved as a plant hormone for stress adaptation, beginning with the emergence of vascular plants.

## 8. Conclusions and Future Perspectives

JA and its derivatives play crucial roles in the defense and resistance of plants in response to biotic and abiotic stresses. The roles of JAs in the plant defense responses and in growth protection provide a direct way of alleviating the stresses. In the presence of abiotic stresses, JAs induce tolerance chiefly by activating the plant’s defense mechanisms, which mainly involve the antioxidative enzymes and other defensive compounds. Future studies will pinpoint how different environmental signals are perceived by plants in the various components in the signaling pathways and the biosynthesis of the JAs, especially in the initiation and establishment of cooperation between the TFs and JAZ repressors during JA signal transduction. Future studies will also elucidate the molecular mechanisms of JA movement through the transporter, resource allocation between growth- and defense-related processes, synergistic or antagonistic interactions between JA and other hormonal signaling pathways. Such works will expand our understanding of the molecular mechanisms underlying the actions of JA against biotic and abiotic stresses.

## Figures and Tables

**Figure 1 ijms-21-00621-f001:**
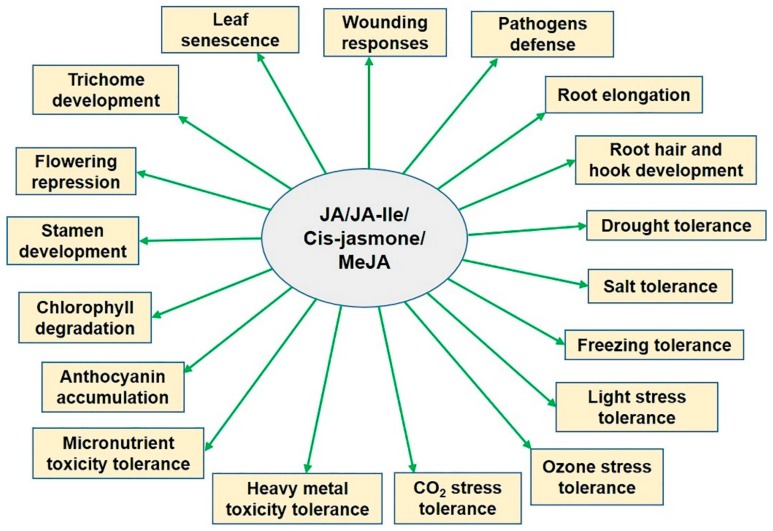
Various plant processes modulated by jasmonic acid and its isoleucine conjugate in response to abiotic stresses. JA, jasmonic acid; JA-Ile, jasmonyl isoleucine; MeJA, methyl jasmonate.

**Figure 2 ijms-21-00621-f002:**
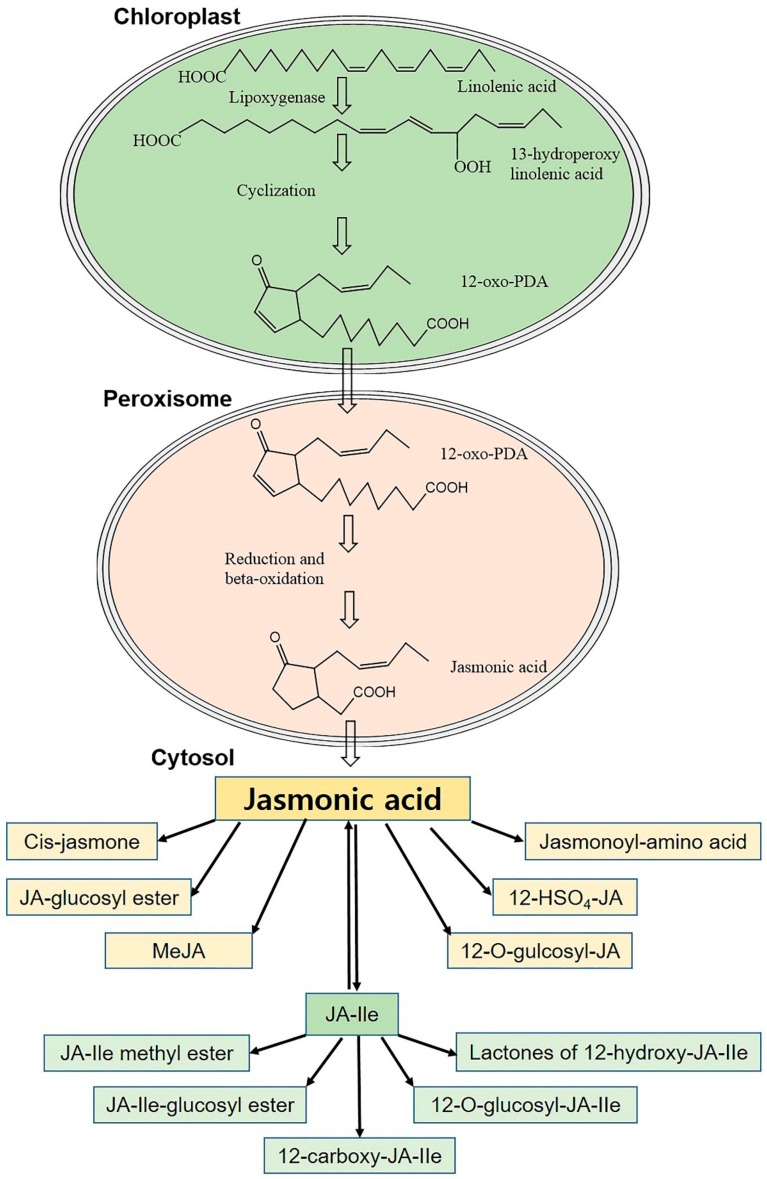
Schematic diagram of jasmonic acid biosynthesis and metabolism in response to abiotic stresses. In the chloroplast, JA biosynthesis begins with the chloroplast membrane release of linolenic acid, which is finally converted to 12-oxo-PDA. Upon transport of 12-oxo-PDA into the peroxisome, a series of enzymes work to convert it to JA, which is then exported to the cytoplasm. JA may be metabolized into different compounds depending on the chemical modification of the carboxylic acid group, the pentenyl side chain, or the pentanone ring. JA, jasmonic acid; JA-Ile, jasmonyl isoleucine; MeJA, methyl jasmonate; 12-HSO_4_-JA, 12-hydroxyjasmonic acid sulfate; 12-oxo-PDA, 12-oxo-phytodienoic acid.

**Figure 3 ijms-21-00621-f003:**
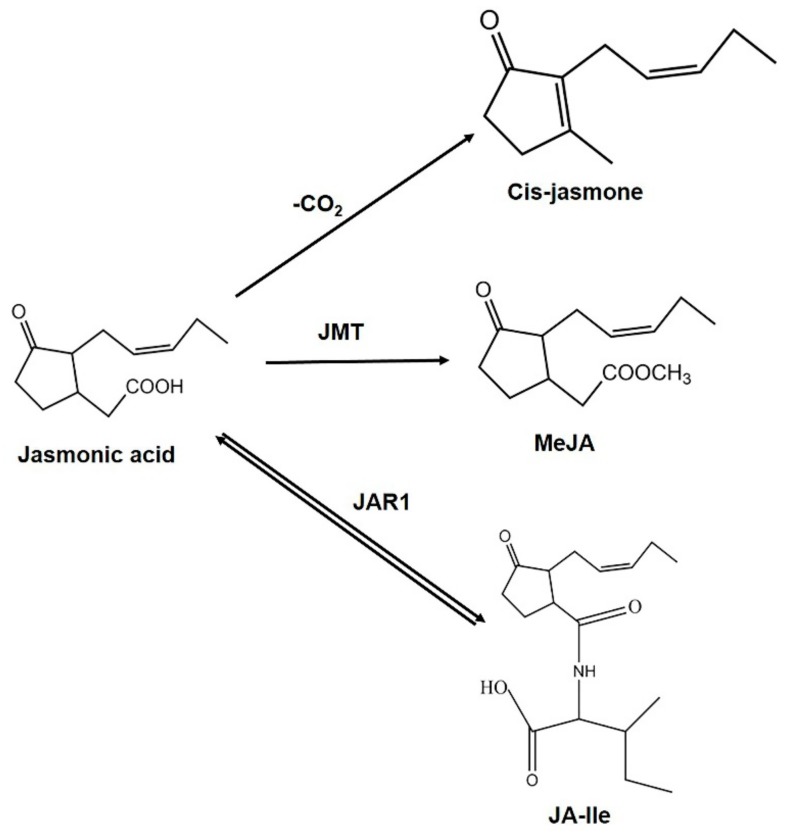
Major bioactive jasmonates in plants and their bioconversion. -CO_2_, decar-boxylation; JMT, jasmonic acid carboxyl methyltransferase; MeJA, methyl jasmonate; JAR1, jasmonate amino acid synthetase 1; JA-Ile, jasmonyl isoleucine.

**Figure 4 ijms-21-00621-f004:**
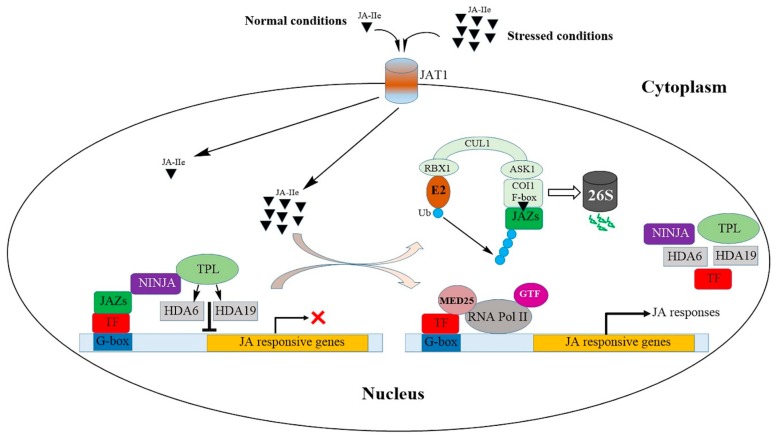
Jasmonic acid perception and signal transduction during abiotic stress. In the absence of abiotic stimuli or at a low level of JA-Ile, the transcription factors are repressed by JAZ proteins, thereby preventing their activation of the promoters of jasmonate-responsive genes. JAZ proteins recruit TPL and adaptor protein NINJA to form an active transcriptional repression complex that inhibits JA responses by changing the open complex to a closed one through the further recruitment of HDA6 and HDA19. Abiotic stresses elevate JA synthesis, which is readily epimerized to JA-Ile. The latter is then transported to the nucleus by the JAT1 transporter. JA-Ile facilitates the interaction of JAZ with the F-box protein COI1 within the SCF complex, leading to the proteasomal degradation of JAZ. The derepressed TF binds to the G-box element, whereupon MED25, RNA Pol II, and GTF are recruited, resulting in the expression of jasmonate-responsive genes. JA, jasmonic acid; JA-Ile, jasmonyl isoleucine; JAT1, jasmonic acid transfer protein 1; TF, transcription factor; JAZ, jasmonate ZIM domain; NINJA, novel interactor of JAZ; TPL, topless; HDA6, HDA19, histone deacetylase 6, 19; Ub, ubiquitin; E2, ubiquitin-conjugating enzymes; RBX1, ring box 1; CUL1, cullin 1; ASK1, *Arabidopsis* SKP1 homolog 1; COI1, coronatine insensitive 1; MED25, mediator 25; RNA Pol II, RNA polymerase II; GTF, general transcription factor.

**Table 1 ijms-21-00621-t001:** Transcription factors that interact with the jasmonate-ZIM domain proteins and their corresponding JA-regulated plant responses (adapted from Zhai et al. [72]; Zhu and Lee [73]).

JAZ Domains	JAZ-Interacting DNA-Binding Transcription Factors	Physiological Functions
JAZs	MYC2/3/4/5	Root elongation, wounding responses, defense, metabolism, hook development [58,74,75,76,77]
JAZ1/8/10/11	MYB21/24	Stamen development and fertility [71]
JAZ1/2/5/6/8/9/10/11	TT8/GL3/EGL3 /MYB75/GL1	Trichome development and anthocyanin synthesis [70]
JAZ1/3/4/9	FIL/YAB1	Chlorophyll degradation and anthocyanin accumulation [78]
JAZ9/11	OsRSS3/OsbHLH148	Confer drought and salt tolerance [79,80]
JAZ1/4/9	ICE1/2	Increase freezing tolerance [68]
JAZ4/8	WRKY57	Promote leaf senescence [69]
JAZ1/3/9	EIN3/EIL1	Root elongation, defense, root hair and hook development [81]
JAZ1/3/4/9	TOE1/2	Repression of flowering during early vegetative development [82]
JAZs except JAZ7/12	bHLH03/13/14/17	Root elongation, fertility, defense, anthocyanin synthesis [83,84,85,86]

JA, jasmonic acid; JAZ, jasmonate ZIM domain.

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
