# Peer review of "Jasmonic Acid Signaling Pathway in Response to Abiotic Stresses in Plants"

_ijms, 2020, doi:10.3390/ijms21020621_

Round 1
Reviewer 1 Report
The review manuscript by Ali and Baek is a compilation of information on JAs- biosynthesis, signaling, regulation, and roles of JA pathway in response to abiotic stresses. Few suggestions are as below:
Since the review is focused on the roles of JA pathways it is a good opportunity for the authors to make it clear somewhere in the Introduction section about why and how their review is focused on “Roles” of JA because sometime this subject is confused by “functions” of JA. If they are using the words role and function interchangeably then they should clear it in the text with an explanation. However, the title of the review is focused on Roles of JA pathway, but this topic is overshadowed by other information such as: JAs – biosynthesis, signaling, and regulation. I would suggest that authors should elaborate the section 6 by adding more insights from the literature and that flows with the title of the manuscript. Section 6 is about roles of JA in alleviating abiotic stresses, but the subsections are mostly on “effects of….” rather “role of JA”. So, I authors should modify this section. The “conclusion and future perspectives” section is written in a very limited way. Authors should elaborate it further by identifying gaps in the scientific knowledge on JA’s roles and how that knowledge will benefit plant and agricultural science.Author Response
Author’s response to reviewers’ comments
We appreciate the reviewers’ comments. We prepared our answers for the comments from the reviewers one-by-one. Our answers for the comments are as follows;
Comments and Suggestions for Authors
The review manuscript by Ali and Baek is a compilation of information on JAs- biosynthesis, signaling, regulation, and roles of JA pathway in response to abiotic stresses. Few suggestions are as below:
Since the review is focused on the roles of JA pathways it is a good opportunity for the authors to make it clear somewhere in the Introduction section about why and how their review is focused on “Roles” of JA because sometime this subject is confused by “functions” of JA. If they are using the words role and function interchangeably then they should clear it in the text with an explanation. However, the title of the review is focused on Roles of JA pathway, but this topic is overshadowed by other information such as: JAs – biosynthesis, signaling, and regulation. I would suggest that authors should elaborate the section 6 by adding more insights from the literature and that flows with the title of the manuscript. Section 6 is about roles of JA in alleviating abiotic stresses, but the subsections are mostly on “effects of….” rather “role of JA”. So, I authors should modify this section. The “conclusion and future perspectives” section is written in a very limited way. Authors should elaborate it further by identifying gaps in the scientific knowledge on JA’s roles and how that knowledge will benefit plant and agricultural science.
Response: We appreciate the reviewer’s comments. We modified the title of our review considering the suggestions of all reviewers.
We also elaborated the section 6 by adding some text as-
“Transcript profile analysis of stressed sweet potato revealed that during salt stress JA level was significantly increased to cope with the effect of salt stress [113].” Added under the subheading of “6.1. Jasmonic Acid Signaling under Salt Stress”
“JAs also play a crucial role in plant defense responses against lead (Pb) stress. JA showed reduction in Pb uptake and increased the growth of tomato plants when seeds were primed with JA [136].” Added under the subheading of “6.4. Jasmonic Acid Signaling under Micronutrient Toxicity”
“Elevated ozone activated the JA pathway in tomato which significantly up-regulated the emission rates of volatile compounds for the protection plants from natural enemies [150].” Added under the subheading of “6.6. Jasmonic Acid Signaling under Ozone Stress”.
We added a subheading as-
6.8. Jasmonic Acid Signaling under CO2 Stress
There are few reports about the JA signal transduction pathway in plants under CO2 stress, however, these reports have varied for various plant and insect species [155–157]. Ballhorn et al. reported that in lima bean (Phaseolus lunatus), the concentration of MeJA and cis-JA was increased at a high concentration of CO2 (500, 700, and 1000 ppm) [158]. An elevated level of CO2 (750 ppm) increased the defense mechanism of tomato plants against nematode by activating the JA- and SA-signaling pathway [159]. The elevated level of CO2 also increased the JA and main defense-related metabolites in tobacco but decreased in rice [157].
We added some text “Future studies will also elucidate the molecular mechanisms of JA movement through the transporter, resource allocation between growth- and defense-related processes, synergistic or antagonistic interactions between JA and other hormonal signaling pathways” in the “Conclusions and Future Perspectives” according to the reviewer’s suggestion.
Reviewer 2 Report
Title needs improvement
I suggest to add Abiotic stresses in plants
“Role of jasmonic acid” – this looks very broad, better to have more precise and informative.
Abstract
The abstract is too short that the usual abstract in the journal. Similarly, only four keywords are provided
“Plants live as immovable organisms in environments that are characterized by the existence of biotic and abiotic stresses.” – This is very generalized and not enough informative to add in the abstract. I suggest removing the sentence.
Overall abstract needs significant improvement. Sentences are not well connected. The authors need to maintain the flow of information.
Introduction
“Plants grow in environments that contain a variety of biotic and abiotic stresses.” – Better to use impose instead of contain.
“For example, extremely high or low temperatures might inhibit a subset of enzymes without affecting other enzymes in the same or connected pathways [3],” - without affecting other enzymes is redundant to a subset of enzymes.
Figure 1 is not informative. Authors have also added developmental stages which are natural processes and not stress, secondly the figure need elaborated information about what key role JA perform in each of the stress provided in the figure.
“The volatile MeJA is formed from JA through the activity of JA carboxyl methyltransferase (Figure 3) [26].”- Term instead of “formed from” can be used
Under normal conditions, the level of JA-Ile is low and thus the promoters of jasmonate-responsive genes are not activated by the different types of transcription factors (TFs) – consider rephrasing
Improve subheadings, for instance, Effect of drought on JA signaling should be – Role of JA signaling under drought stress or JA signaling under drought stress.
“Exogenous MeJA increased the resistance of the black locust tree (Robinia pseudoacacia) to salt stress by increasing the activities of superoxide dismutase (SOD) and ascorbate peroxidase (APX) [95].”- Word resistance can be substituted with tolerance.
“MeJA mitigated the toxicity of boron in the sweet woodworm (Artemisia annua) by reducing the amount of lipid peroxidation and stimulating the synthesis of antioxidative enzymes [115].”- Correct the name sweet woodworm to sweet wormwood
“The oxidative stress caused by arsenic in B. napus was minimized by MeJA through its induction of the expression of genes coding for antioxidants and secondary metabolites [116].”- The sentence appears confusing, consider paraphrasing
Conclusion
“In the presence of abiotic stresses, JAs induce resistance chiefly by activating the plant`s defense mechanisms, which mainly involve the antioxidative enzymes and other defensive compounds.”- Use term tolerance instead of resistance
Author Response
Author’s response to reviewers’ comments
We appreciate the reviewers’ comments. We prepared our answers for the comments from the reviewers one-by-one. Our answers for the comments are as follows;
Comments and Suggestions for Authors
Title needs improvement
I suggest to add Abiotic stresses in plants
“Role of jasmonic acid” – this looks very broad, better to have more precise and informative.
Response: We appreciate the reviewer’s comments. We changed the title as “Jasmonic Acid Signaling Pathway in Response to Abiotic Stresses in Plants”.
Abstract
Comment: The abstract is too short that the usual abstract in the journal. Similarly, only four keywords are provided.
Response: We modified the abstract and also added some key words as the reviewers suggested.
Comment: “Plants live as immovable organisms in environments that are characterized by the existence of biotic and abiotic stresses.” – This is very generalized and not enough informative to add in the abstract. I suggest removing the sentence.
Response: We removed the sentence as reviewer’s suggested.
Comment: Overall abstract needs significant improvement. Sentences are not well connected. The authors need to maintain the flow of information.
Response: We change a lot in the abstract and maintained the flow of information within the sentences.
Introduction
Comment: “Plants grow in environments that contain a variety of biotic and abiotic stresses.” – Better to use impose instead of contain.
Response: We replaced the word “contain” by “impose”.
Comment: “For example, extremely high or low temperatures might inhibit a subset of enzymes without affecting other enzymes in the same or connected pathways [3],” - without affecting other enzymes is redundant to a subset of enzymes.
Response: We changed the sentence as “For example, extremely high or low temperatures might inhibit a subset of enzymes in the same or connected pathways [3], and hence various intermediate compounds might accumulate as a result of this functional uncoupling of metabolic pathways [4].”
Comment: Figure 1 is not informative. Authors have also added developmental stages which are natural processes and not stress, secondly the figure need elaborated information about what key role JA perform in each of the stress provided in the figure.
Response: We appreciate the reviewer for his critical reviewing. We changed the figure and included only the JA stress related topics.
Comment: “The volatile MeJA is formed from JA through the activity of JA carboxyl methyltransferase (Figure 3) [26].”- Term instead of “formed from”, can be used.
Response: We changed the sentence as “The volatile MeJA is produced from JA through the activity of JA carboxyl methyltransferase (Figure 3) [26].”
Comment: Under normal conditions, the level of JA-Ile is low and thus the promoters of jasmonate-responsive genes are not activated by the different types of transcription factors (TFs) – consider rephrasing.
Response: We changed the sentence as “Under normal conditions, the promoters of jasmonate-responsive genes are not activated by the different types of transcription factors (TFs) due to the low level of JA-Ile (Figure 4).”
Comment: Improve subheadings, for instance, Effect of drought on JA signaling should be – Role of JA signaling under drought stress or JA signaling under drought stress.
Response: We appreciate the reviewer’s kind suggestions. We changed all the subheading as reviewer suggested.
Comment: “Exogenous MeJA increased the resistance of the black locust tree (Robinia pseudoacacia) to salt stress by increasing the activities of superoxide dismutase (SOD) and ascorbate peroxidase (APX) [95].”- Word resistance can be substituted with tolerance.
Response: We substituted the word “Resistance” by “Tolerance”.
Comment: “MeJA mitigated the toxicity of boron in the sweet woodworm (Artemisia annua) by reducing the amount of lipid peroxidation and stimulating the synthesis of antioxidative enzymes [115].”- Correct the name sweet woodworm to sweet wormwood.
Response: We corrected the name.
Comment: “The oxidative stress caused by arsenic in B. napus was minimized by MeJA through its induction of the expression of genes coding for antioxidants and secondary metabolites [116].”- The sentence appears confusing, consider paraphrasing.
Response: We paraphrased the sentence as “In B. napus, oxidative stress minimized by MeJA through the induction of the expression of genes encoding antioxidants and secondary metabolites.”
Conclusion
Comment: “In the presence of abiotic stresses, JAs induce resistance chiefly by activating the plant`s defense mechanisms, which mainly involve the antioxidative enzymes and other defensive compounds.”- Use term tolerance instead of resistance.
Response: We substituted the word “resistance” by “tolerance”.
Reviewer 3 Report
The manuscript by Ali and Saekaims to review the roles of Jasmonates under abiotic stress in Plants. Unfortunately, many major issues do not allow to this paper to reach the minimum standards required for a renowned international journal as IJMS.
This is not what a review paper should be. The manuscript looks like a list of findings about very well-known notions about roles, synthesis and effects of JA in plants under stress.
The involvement of JA in abiotic stress has been recently investigated and reviewed in a number of papers (many from IJMS and MDPI), thus a strong background and original view on this topic is strongly required, in order to involve Readers in consulting the review.
On the contrary, the Authors do not seem to possess the necessary background to properly develop a review on this topic; as confirm, both of them have limited experience in JA studies, generally confined in the pathogen response field.
As consequence, the whole matter is described in a highly generic way, and the manuscript does not focus on a specific topic: the JA occurrence, signaling and effects are described generally, with the result that the paper is full of very well-known notions, and it is too scholastic in many points.
No clear, specific and new conclusion is present, and it is not clear where the originality of this review lays.
It looks more as an Intro of a PhD thesis that a review for a high rank journal such as IJMS.
Therefore, I regret to suggest the rejection of the manuscript for publication on IJMS.
Author Response
Author’s response to reviewers’ comments
We appreciate the reviewers’ comments. We prepared our answers for the comments from the reviewers one-by-one. Our answers for the comments are as follows;
Comments and Suggestions for Authors
The manuscript by Ali and Baek aims to review the roles of Jasmonates under abiotic stress in Plants. Unfortunately, many major issues do not allow to this paper to reach the minimum standards required for a renowned international journal as IJMS.
This is not what a review paper should be. The manuscript looks like a list of findings about very well-known notions about roles, synthesis and effects of JA in plants under stress.
The involvement of JA in abiotic stress has been recently investigated and reviewed in a number of papers (many from IJMS and MDPI), thus a strong background and original view on this topic is strongly required, in order to involve Readers in consulting the review.
On the contrary, the Authors do not seem to possess the necessary background to properly develop a review on this topic; as confirm, both of them have limited experience in JA studies, generally confined in the pathogen response field.
As consequence, the whole matter is described in a highly generic way, and the manuscript does not focus on a specific topic: the JA occurrence, signaling and effects are described generally, with the result that the paper is full of very well-known notions, and it is too scholastic in many points.
No clear, specific and new conclusion is present, and it is not clear where the originality of this review lays.
It looks more as an Intro of a PhD thesis that a review for a high rank journal such as IJMS.
Therefore, I regret to suggest the rejection of the manuscript for publication on IJMS.
Response: We appreciate the reviewer’s critical comments. We changed a lot in our manuscript according to the comments given by the other three reviewers. Hope the modified version of our review will impress the reviewer.
Reviewer 4 Report
In the review article entitled “Roles of the Jasmonic Acid Pathway in Response to Abiotic Stresses”, the authors give a comprehensive review of the biosynthesis and signal transduction pathways of the Jasmonic Acids(JAs) and the roles of these molecules in the plant responses to abiotic stresses. Overall, the review is well-structured and can be served as a starting point for studies that aim to expand the understanding of the molecular mechanisms underlying the actions of JA against biotic and abiotic stresses. I have no major concerns. However, the manuscript can be improved if the authors can address the following minor points:
1, Line 104, the authors cite ref 43 as evidence to support that “JA-Ile is an important compound in the JA signal transduction pathway”. However, it seems to me that ref 43 is not a proper paper to support this. The authors may cite the wrong reference.
2, Line 124, it seems that the ref 45 (“SlMAPK3 enhances tolerance to tomato yellow leaf curl virus (TYLCV) by regulating salicylic acid and jasmonic acid signaling in tomato (Solanum lycopersicum)”) is not the right reference.
3, Line 126, after “and apoplast.”, a proper reference is needed.
4, Line 191-194, proper references are needed.
5, for Table 1, it will be appreciated that the authors add references for each row.
Author Response
Author’s response to reviewers’ comments
We appreciate the reviewers’ comments. We prepared our answers for the comments from the reviewers one-by-one. Our answers for the comments are as follows;
Comments and Suggestions for Authors
In the review article entitled “Roles of the Jasmonic Acid Pathway in Response to Abiotic Stresses”, the authors give a comprehensive review of the biosynthesis and signal transduction pathways of the Jasmonic Acids (JAs) and the roles of these molecules in the plant responses to abiotic stresses. Overall, the review is well-structured and can be served as a starting point for studies that aim to expand the understanding of the molecular mechanisms underlying the actions of JA against biotic and abiotic stresses. I have no major concerns. However, the manuscript can be improved if the authors can address the following minor points:
Comment 1, Line 104, the authors cite ref 43 as evidence to support that “JA-Ile is an important compound in the JA signal transduction pathway”. However, it seems to me that ref 43 is not a proper paper to support this. The authors may cite the wrong reference.
Response: We appreciate the reviewer comment. We changed the reference by proper reference.
Comment 2, Line 124, it seems that the ref 45 (“SlMAPK3 enhances tolerance to tomato yellow leaf curl virus (TYLCV) by regulating salicylic acid and jasmonic acid signaling in tomato (Solanum lycopersicum)”) is not the right reference.
Response: We changed the reference by proper reference.
Comment 3, Line 126, after “and apoplast.”, a proper reference is needed.
Response: We added reference as reviewer suggested.
Comment 4, Line 191-194, proper references are needed.
Response: We added references as reviewer suggested.
Comment 5, for Table 1, it will be appreciated that the authors add references for each row.
Response: We rearranged the table and also added references for each row.
Round 2
Reviewer 1 Report
The authors have done a great job in improving the manuscript significantly, but the section 6 still need further improvements. The authors have put the section 6 on “roles played by JA in alleviating the effects of abiotic stresses”. Considering the heading it would be nice if the authors start with explaining stress effects and then explaining it further what role JA play that leads to alleviation of those effects. Following is an example specifically for section 6.1, but the authors can apply a similar approach to improve the sections from 6.2 to 6.8:
Suggestion on section 6.1: The authors are giving various examples where JA content was shown to be increased in plants. It would be better if authors provide few examples – what role the increased JA content played that led to the lessening of stress effects, e.g. L265 can be elaborated by putting more information on how increased SOD and APX, which was increased due to the increase in MeJA, helped to tolerate salt stress. Similarly, L267-268 where authors write that “JAs can alleviate salt stress by increasing the endogenous hormones”. I would suggest elaborating it by giving few specific examples of hormones that were found to be increased as a result of increased JA content in plant cell and their respective roles in improving stress tolerance.
Reviewer 2 Report
The authors have addressed all of my concerns. The revised version of the MS looks appropriate.
Reviewer 3 Report
Th Authors did not amend the manuscript as requested in the previous review. They only change 3-4 sentences, and the whole work is still very far from the standards required.
Therefore, I regret to suggest the reject of the manuscript for publication on IJMS.